# The Role of Gut Microbiota in Orthopedic Surgery: A Systematic Review

**DOI:** 10.3390/microorganisms13051048

**Published:** 2025-04-30

**Authors:** Ahmed Nadeem-Tariq, Sarah Kazemeini, Matthew Michelberger, Christopher Fang, Sukanta Maitra, Karen Nelson

**Affiliations:** Kirk Kerkorian School of Medicine at UNLV, 625 Shadow Lane, Las Vegas, NV 89106, USA

**Keywords:** gut microbiota, microbiome, orthopedic surgery

## Abstract

The human gut microbiome represents a complex ecosystem comprising numerous microorganisms critical to basic physiological processes. The gut microbiome’s composition and functionality influence surgical outcomes following orthopedic procedures. The purpose of this study was to evaluate the gut microbiota on critical aspects of orthopedic surgical outcomes. A comprehensive literature search was conducted via PubMed, the Cumulative Index for Nursing and Allied Health Literature (CINAHL), Google Scholar, Cochrane Library, Medline, and Web of Science. A total of 18 research articles of the 599 retrieved results were included in this study. Significant correlations were identified between microbial composition and surgical outcomes, including infection rates, inflammatory responses, and postoperative complications. Bacterial genera like *Alistipes* and *Helicobacter* increased postoperative cognitive dysfunction (POCD) risk, while short-chain fatty acid (SCFA)-producing bacteria showed negative correlations with inflammatory markers. Probiotic interventions reduced POCD incidence from 16.4% to 5.1% and modulated inflammatory responses. Additionally, bacterial composition was associated with critical surgical parameters such as bone healing, infection rate, and recovery trajectory. Inflammation, healing processes, and recovery trajectories are influenced by microbial composition in surgical settings. Targeted interventions, such as probiotics, show promise in reducing surgical risks and improving patient recovery.

## 1. Introduction

The gut microbiota is a complex and dynamic ecosystem consisting of bacteria, viruses, fungi, and protozoa that reside within the gastrointestinal tract [1,2]. It comprises over 1000 species encoding more than three million genes, and its composition varies significantly between individuals [3,4]. Most microorganisms are concentrated in the large intestine, where they perform critical functions in maintaining local intestinal barrier integrity and modulating systemic immune responses [5,6,7]. Disruptions in its diversity and composition have been linked to numerous local and systemic diseases [8].

Gut microbiota evolves in response to host genetic makeup, dietary habits, lifestyle, environment, and medical history [9]. They are involved in various basic physiological aspects, such as the metabolism of nutrition, synthesis of important vitamins, maintenance of the immune system, modulation of inflammation, and protection against pathogenic microorganisms [10,11,12]. Disruptions to this balance, known as dysbiosis, have been linked to inflammatory bowel diseases, metabolic disorders, and impaired immune responses [13]. Microbial dysbiosis may correlate with compromised bone mineral density, increased infection risks in joint replacements, and altered healing mechanisms in fracture repair [14]. Specific bacterial populations have been associated with enhanced or diminished bone regenerative capacities, highlighting potential therapeutic opportunities through microbiome modulation [15].

The intricate relationship between the microbiome, host immunity, and inflammatory regulation emphasizes its critical role in surgical contexts. Surgical interventions disrupt the microbiome’s balance by triggering inflammatory cascades and metabolic changes [16,17,18]. Perioperative factors such as antibiotics and nutritional deprivation can alter the gut microbiome’s diversity, leading to surgical site infections [19,20]. Microbiome alterations significantly impact clinical outcomes in orthopedic surgeries by increasing infection risk and hindering recovery [21]. Joint replacements and fracture repairs are among the most commonly performed procedures worldwide, but infection and delayed healing remain as the leading causes of postoperative complications. The invasive nature of these surgeries and their direct impact on musculoskeletal tissues underscore the urgent need to understand and manage microbiome alterations to optimize patient outcomes [22,23].

Despite the growing interest in the microbiome’s role in inflammatory and surgical contexts, significant gaps in the literature persist. These include the mechanisms behind microbiome-mediated inflammatory responses, long-term impacts of surgical interventions on microbial composition, and variability in microbiome responses across different orthopedic procedures. There is also a need to identify predictive microbiological markers and develop interventional strategies leveraging microbiome insights to improve orthopedic surgical outcomes.

Beyond orthopedics, microbiome testing is of particular interest in specialties such as general surgery, where gut microbial composition has been linked to complications such as anastomotic leaks, malabsorption, ileus, and surgical site infections [24]. Preoperative microbiome profiling offers potential for risk stratification and personalized perioperative management, allowing clinicians to predict and mitigate adverse outcomes more effectively.

Despite its promise, widespread implementation of microbiome diagnostics remains limited by the lack of standardized clinical guidelines and inconsistent insurance coverage. However, recent international efforts to establish regulatory frameworks for microbiome testing may pave the way for broader clinical adoption and reimbursement [25]. Expanding research in this space could shape future surgical guidelines and improve access to microbiome-informed care.

This study summarizes the current text available on the associations between gut microbiota and orthopedic surgical outcomes. It aims to synthesize current evidence on the composition of gut microbiota and its influence on recovery following orthopedic procedures. Furthermore, it will evaluate the relationships between microbial dynamics and key surgical outcomes, identify potential microbiological markers predictive of surgical success, and investigate therapeutic interventions targeting microbiome modulation. This review will provide actionable insights for improving patient outcomes in orthopedic surgery through microbiome-informed strategies.

Research Objectives: The purpose of this study was to comprehensively explore how gut microbiota influences surgical outcomes following orthopedic procedures, and investigate the impact of gut microbiota on postoperative infection rates, bone healing, and recovery.

Research Questions: How do the composition and diversity of gut microbiota influence postoperative outcomes in orthopedic surgery? How do probiotic and prebiotic interventions modulate gut microbiota to influence surgical outcomes in orthopedic surgery?

## 2. Methodology

The reporting of this study adhered to the Preferred Reporting Items for Systematic Reviews and Meta-Analysis (PRISMA) [26]. For PubMed, search terms included (“gut microbiota” OR “intestinal microbiota” OR “microbiome”) AND (“orthopedic surgery” OR “joint replacement”). Filters for English language peer-reviewed articles were applied. For the Cumulative Index for Nursing and Allied Health Literature (CINAHL), combinations such as “intestinal bacteria AND arthroplasty” were used. Google Scholar searches incorporated broader terms with Boolean operators to include gray literature. Cochrane Library searches focused on randomized controlled trials, using the string “microbiome AND orthopedic outcomes.” Medline and Web of Science searches followed similar patterns, emphasizing human subjects and excluding conference abstracts or non-original research. ChatGPT-4o was used to refine certain ideas within the key takeaways for orthopedic surgeons section to condense and summarize key findings in a practical manner.

### 2.1. Identification and Selection of Studies

A comprehensive literature search for peer-reviewed original research articles was conducted via PubMed, the CINAHL, Cochrane Library, Medline, and Web of Science.

### 2.2. Search Strategy

Keywords were identified and used with MeSH terms to formulate search strings. The following keywords were used in different combinations to optimize the search: (“gut microbiota” OR “intestinal microbiota” OR “microbiome” OR “gut flora” OR “intestinal bacteria”) AND (“orthopedic surgery” OR “orthopedic procedures” OR “musculoskeletal surgery” OR “joint replacement” OR “arthroplasty” OR “fracture fixation”).

### 2.3. Study Selection

A two-reviewer system was used in the screening and article selection for consistency. Discrepancies were resolved through discussion or the intervention of a third reviewer. The retrieved results from various databases were exported to Zotero screening software version 6.0.36. The software automatically detected retracted articles and identified duplicate records, which a reviewer manually merged. Non-duplicate records were then screened using prespecified eligibility criteria.

#### 2.3.1. Eligibility Criteria

This study included research on the role of gut microbiota in orthopedic surgery. Articles fulfilling the modified population intervention comparison primary outcomes study design (PICOS) criteria were selected [25].

The PICOS criteria for eligible studies were defined as follows;

Population (P): patients undergoing orthopedic surgeries.Intervention (I): gut microbiota.Comparison (C): placebo groups.Primary outcomes (O): pinfection rates, bone healing, microbiota composition changes, incidence of postoperative cognitive dysfunction, and bone mineral density.Study Design (S): randomized controlled trials (RCTs), prospective and retrospective cohort studies, case-control studies, observational cohort studies, and any other suitable study designs.

The potential articles were subjected to the following inclusion and exclusion criteria:

#### 2.3.2. Inclusion Criteria

This study included peer-reviewed original research articles published in English. It included studies on human subjects only undergoing orthopedic procedures. In addition, research articles investigating the association between gut microbiota, inflammation markers, bacterial endotoxins, and various clinical outcomes were included.

#### 2.3.3. Exclusion Criteria

Study protocols, reviews, meta-analyses, letters, editorials, conference abstracts, and opinion pieces were excluded.

### 2.4. Methodological Quality Assessment

The risk of bias visualization tool developed by the Cochrane Collaboration (Rob 2.0) was used to assess the risk of bias in the eligible RCTs. The tool assessed bias arising from the randomization process, deviation from the intended intervention, missing outcome data, measurement of outcome, and selection of reported results [27]. In addition, the eligible cohort studies’ risk of bias was assessed using the Newcastle Ottawa Scale (NOS) evaluating the following domains: the selection of the study groups, the comparability of the groups, and the ascertainment outcome of interest. The National Institutes of Health (NIH) quality assessment tool was used to assess the risk of bias in the eligible case-control studies [28].

### 2.5. Data Selection and Extraction

Data from the included studies were systematically extracted and tabulated using Microsoft Excel 2024. The following data sets were extracted: authors; study region; sample size; demographics; study design; intervention; primary outcomes: infection rates, bone healing; and secondary outcomes: pain, inflammation, hospital stay, objectives, and findings.

### 2.6. Data Analysis

The extracted data from the studies were thematically analyzed according to the outcome measures of interest by critically reading the data and noting down the initial observations while procedurally coding and grouping them into potential themes. The coded themes were then checked for accuracy [29].

## 3. Results

### 3.1. Study Selection

The database search yielded 599 records, of which 78 duplicates were removed. Further, 474 articles were excluded following the title and abstract screening. The remaining 47 articles were sought for retrieval, after which two RCTs, three case-control, and 12 cohort studies that met the eligibility criteria were included. The results are presented in Figure 1.

### 3.2. Methodological Quality Assessment

There was a low overall risk of bias in the included RCTs, as shown in Figure 2 and Figure 3. In addition, the included cohort studies demonstrated good quality, as shown in Table 1. Moreover, the case-control studies included had good overall quality, as shown in Table 2. Table 3 provides a summary of the studies after quality assessment.

Figure 2 presents a traffic light plot summarizing the risk of bias assessments for each included study using the Cochrane Risk of Bias 2.0 (RoB 2.0) tool. Each row corresponds to an individual randomized controlled trial, while the columns represent five specific domains of bias: bias arising from the randomization process, bias due to deviations from intended interventions, bias due to missing outcome data, bias in measurement of the outcome, and bias in the selection of the reported result. Color coding is used to visually represent the level of concern within each domain: green indicates a low risk of bias, yellow indicates some concerns, and red indicates a high risk. All three studies were assessed as having an overall low risk of bias [30,31,32].

Figure 3 provides a summary plot that aggregates the domain-specific risk of bias judgments across all included studies. Each bar illustrates the proportion of studies assessed as low risk or having some concerns for each of the five RoB 2.0 domains. The majority of assessments fell into the low-risk category, reflecting overall strong methodological quality across the studies. Notably, the domain concerning missing outcome data was the only category with a study assessed as having some concerns, accounting for approximately one-third of the total sample. This overview reinforces the reliability of the evidence base included in this systematic review while highlighting a specific area that warrants attention in future trials.

### 3.3. Data Selection and Extraction

The included studies focused on orthopedic and neurological patient populations from the USA, China, Ireland, Switzerland, Korea, and Spain. Most studies utilized advanced microbiological techniques such as 16S rRNA sequencing to analyze gut microbiota composition and its implications for surgical outcomes. The sample sizes ranged from 21 to 551 patients. The studies employed prospective and retrospective cohorts, randomized controlled trials, and case-control research designs.

The studies investigated postoperative cognitive dysfunction, periprosthetic joint infections, bone healing, and neurological outcomes. In addition, they investigated relationships between gut microbiota, inflammation markers, bacterial endotoxins, and clinical outcomes such as infection rates, pain, cognitive function, and bone mineral density. The studies used microbiome analysis, probiotic interventions, metabolite measurements, and biomarker assessments in the analyses.

### 3.4. Thematic Analysis of Outcomes

#### 3.4.1. Gut Microbiota and Surgical Outcomes

The surgical outcomes for procedures such as spinal fusion and joint replacement are significantly influenced by gut microbiota. This is particularly evident in cases involving surgical site infection (SSI) and wound healing. Dysbiotic microbiota, characterized by an over-abundance of genera such as Alistipes (*p* = 0.008) and Helicobacter (*p* = 0.007), was associated with postoperative cognitive dysfunction (POCD). [29] Pathways like glycosaminoglycan biosynthesis were associated with tissue regeneration, suggesting the importance of gut microbiota in recovery post-surgery. Preoperative gut microbiota composition influences infection rates and recovery trajectories following surgery. Patients with pre-existing dysbiosis, such as low Firmicutes–Bacteroidetes ratio, experience delayed recovery and higher infection rates [30].

Gut microbiota composition also significantly influences bone and soft tissue healing. The changes in gut microbiota were found to be significantly associated with the changes in bone mineral density (BMD), hence affecting bone health. In spinal fusion, significant differences in gut microbiota appear between those with low bone mass (T-score ≤ −1.0) and those with normal bone mineral density (*p* = 0.03) [39]. Das et al. (2019) reported that six genera showed altered abundance in either osteopenia or osteoporosis subjects, five of which remained significant after adjustment for confounders. β-Diversity analysis explained 15–17% of microbiota variance, of which 2% of the variance was contributed by BMD measurements (*p* ≤ 0.05) [43]. Osteoporosis was enriched by the genera *Actinomyces* and *Lactobacillus*, while *Veillonella* was more abundant in osteopenia.

Despite these significant associations, α-diversity metrics did not differ across groups of BMD [35]. However, elevated levels of gut-derived metabolites, such as trimethylamine N-oxide (TMAO), were linked to adverse outcomes, including delayed wound healing. Trimethylamine N-oxide (TMAO) levels correlate positively with systemic inflammatory markers (e.g., IL-6, *p* < 0.01) and oxidative stress (ROS, *p* < 0.001) [31]. Additionally, the Shannon index displayed a lower gut microbiota diversity among the patients in the osteopenia group [33].

#### 3.4.2. Antibiotic Use and Resistance in Orthopedic Surgery

Antibiotic administration is essential in preventing infections during orthopedic surgeries, such as joint arthroplasty and fracture fixation. However, antibiotics can disrupt the delicate balance of the gut microbiota, leading to dysbiosis and systemic complications that affect surgical outcomes. Dysbiosis compromises the gut’s ability to regulate immune responses, increasing susceptibility to opportunistic infections, impaired wound healing, and inflammation. Antibiotic-induced reductions in microbiota diversity diminish the production of short-chain fatty acids (SCFAs). SCFAs are vital for reducing endotoxin translocation, suppressing systemic inflammatory markers like C-reactive protein (CRP), and supporting immune homeostasis [38].

Markers like Zonulin and soluble CD14 (sCD14) provide insight into how gut microbiota disruptions contribute to periprosthetic joint infections (PJIs). Zonulin regulates intestinal tight junctions, but its overexpression compromises gut barrier integrity. Bacterial endotoxins, such as lipopolysaccharides (LPS), enter systemic circulation, and this process triggers systemic inflammation, exacerbates tissue damage, and increases the risk of surgical site infections [34]. Zonulin levels are higher in acute PJIs (10.7 ± 6.2 ng/mL) compared to chronic cases (5.8 ± 4.8 ng/mL, with a *p* = 0.005) [40]. Similarly, sCD14, a co-receptor for endotoxin recognition, is elevated in acute PJIs (555 ± 216 ng/mL, *p* = 0.003) [34,48].

Microbial clustering studies identified differences in bacterial profiles between acute and chronic PJIs. Acute cases often involve *Staphylococcus*-type microbiota (71.4%, *p* = 0.043), whereas chronic PJIs were linked to *Cutibacterium* (11.1%, *p* = 0.036) [35]. Patients with nonunion fractures showed a predominance of gram-positive bacteria (54.3%), but the time to the union did not differ between gram-positive and gram-negative cases (662.3 vs. 446.8 days, *p* = 0.69). However, three gram-positive cases required amputation or arthroplasty before the union [39].

Antibiotic use also influences diagnostic biomarkers in PJIs. Synovial fluid α-defensin exhibited high diagnostic performance as reflected by an area under the curve (AUC) of 0.93, sensitivity of 94.4%, and specificity of 89.5% at a cut-off of 1580 μg/L (*p* < 0.001) [34]. Patients with PJIs also had lower operational taxonomic units in microbiota profiles (133 versus 265, *p* = 0.006), reflecting significant disruptions to microbiota composition [34]. Although biomarkers like α-defensin and leukocyte esterase help identify PJIs, microbiota diversity metrics provide additional insights by highlighting infection-associated microbial changes.

Emerging evidence suggests that supplementing antibiotic regimens with probiotics and prebiotics can restore microbial diversity, support SCFA production, and mitigate inflammatory responses. Strains such as *Lactobacillus acidophilus* and *Bifidobacterium longum* have demonstrated efficacy in reducing CRP levels, lowering infection risks, and promoting recovery [40]. This balanced approach highlights the possibility of integrating microbiota-targeted interventions alongside traditional infection prevention strategies to optimize outcomes in orthopedic surgery. Overall, antibiotic use, though detrimental to gut health, is still important to use to prevent infections.

#### 3.4.3. Probiotic and Prebiotic Interventions

Building on the evidence of gut microbiota’s role in surgical outcomes, exploring the benefits of probiotics and prebiotics provide insights into reducing complications. Probiotic use in surgical patients has shown significant effects in reducing infections and improving recovery. Strains like *Bifidobacterium longum*, *Lactobacillus acidophilus*, and *Streptococcus faecalis* effectively alter the gut microbiota composition and reduce postoperative complications, including cognitive dysfunction (*p* = 0.01). Probiotic administration significantly reduced the incidence of POCD from 16.4% to 5.1% (*p* = 0.046) [37]. Additionally, elevated plasma LPS and CRP levels in poorer outcome surgery patients highlight their potential to be managed or improved by prebiotic treatment [31].

## 4. Discussion

This study hypothesized that gut microbiota composition influences orthopedic surgical outcomes, with variations in microbial diversity and specific bacterial populations correlating with inflammation, infection rates, bone healing processes, and postoperative cognitive function.

The findings revealed multifaceted relationships between gut microbiota and orthopedic surgical outcomes. The study demonstrated that microbiota composition plays a critical role in modulating surgical recovery through complex inflammatory and metabolic mechanisms [30,38]. Probiotic and prebiotic interventions have promising potential in reducing postoperative complications, especially POCD [37].

Bacterial genera, such as *Alistipes* and *Helicobacter*, were associated with increased risks of POCD, while short-chain fatty acids (SCFAs) producing bacteria had inverse associations with inflammatory markers [30]. The composition of the microbiota was different in patients with divergent surgical outcomes, including divergent bone mineral density, susceptibility to infection, and divergent recovery trajectories [39]. In addition, there were postoperative changes in some plasma markers of inflammation, such as CRP and IL-6 [40]. The most dramatic changes in microbiota occurred in those patients who had cognitive dysfunction, highlighting the complex interaction between gut microbes, neurological function, and surgical stress [37,38,40].

Probiotic strains like *Bifidobacterium longum* and *Lactobacillus acidophilus* aid in the postoperative complications and modulation of immune responses [37]. The correlations between gut microbiota and surgical outcomes are complex, multidirectional interactions based on inflammatory pathways, metabolic stress, and immune modulation. Similarly, microbial dysbiosis is an active mediator of the process of surgical recovery apart from a passive marker [38].

The pathways related to glycosaminoglycan biosynthesis point to complex molecular interactions between microbial populations and host tissue recovery [30]. Variations in the composition of microbiota across different patient groups, especially those with cognitive disorders, suggest that surgical stress triggers distinct microbial adaptations [29]. The decline in the Firmicutes–Bacteroidetes ratio in certain patient groups further underscores the dynamic nature of gut microbiome responses to surgical interventions [31].

Malnutrition is a known risk factor associated with orthopedic surgery, which current literature reports as a factor that impedes a patient’s ability to respond to the metabolic demand of surgery. Gut microbiota respond to daily habits such as eating and are involved with nutrition. Disruptions to the microbiota can lead to a delay in immune response [10,11,12,13,49]. Furthermore, nutritional deprivation negatively alters the microbiome, which puts postoperative patients of orthopedic surgery at a higher risk of surgical site infections, renal complications, and mortality [50]. With growing recognition of the role of nutrition in surgical recovery, optimizing key micronutrient levels has become a priority in orthopedic care. Beyond its well-established role in calcium homeostasis and bone mineralization, Vitamin D has been shown to influence gut microbial composition by promoting beneficial bacterial populations and reducing systemic inflammation [51]. The vitamin D receptor, widely expressed in intestinal epithelial cells, is key in maintaining gut barrier integrity and microbial homeostasis [52,53]. Given its relative safety and ease of administration, vitamin D supplementation may be an effective strategy to support both microbiome stability and orthopedic recovery.

Recent literature further supports the critical role of the gut microbiome in skeletal health beyond surgical outcomes. Gut microbes influence bone density, joint integrity, and musculoskeletal resilience through SCFA production, estrogen regulation, and vitamin synthesis [1]. Dysbiosis has been linked to various orthopedic conditions, including osteoarthritis, osteoporosis, intervertebral disc degeneration, neuropathies, and myopathies [47].

In addition to SCFAs and hormonal regulation, gut microbiota contribute to skeletal health through their role in vitamin K metabolism [54]. Certain bacterial strains, including *Escherichia coli* and *Lactobacillus*, are major producers of vitamin K2 in the gut [55]. Bacterially synthesized vitamin K2 is absorbed in the ileum and plays a crucial role in activating vitamin K-dependent proteins, such as osteocalcin and matrix Gla protein, which are essential for bone turnover and vascular calcification prevention [56,57]. Dysbiosis may impair this synthesis, contributing to poor bone mineralization and increased fracture risk [58]. Furthermore, microbiome-mediated modulation of osteoblast and osteoclast activity directly affects fracture healing and bone remodeling [31].

Given the observed relationships between gut microbiota composition and surgical outcomes, targeted interventions such as probiotics and prebiotics offer promising strategies to address complications. The significant reduction in POCD from 16.4% to 5.1% with targeted probiotic use represents a clinically meaningful intervention strategy [44,45]. These findings align with existing literature on microbiome-mediated inflammatory responses. The observed correlations between gut microbiota and bone mineral density expand upon limited existing research, providing a more comprehensive understanding of microbial roles in orthopedic healing processes.

To better visualize the relationship between the gut microbiome and postoperative orthopedic outcomes, Figure 4 presents a side-by-side schematic comparison of healthy versus unhealthy microbial environments and their downstream effects on bone healing, inflammation, and recovery. A balanced gut microbiome—characterized by SCFA-producing species such as *Lactobacillus* and *Bifidobacterium*—promotes anti-inflammatory signaling (e.g., reduced CRP), enhanced vitamin K metabolism, and microbial metabolite production, all of which contribute to improved surgical recovery, lower infection risk, and accelerated bone union. In contrast, dysbiosis—marked by reduced microbial diversity and the presence of species such as *Alistipes* and *Helicobacter*—leads to increased zonulin-mediated intestinal permeability (“leaky gut”), systemic inflammation (elevated IL-6, CRP), and impaired osseointegration. These disruptions contribute to delayed healing, higher periprosthetic joint infection (PJI) risk, and potential neurocognitive decline following surgery. This visual framework reinforces the critical role of microbial composition in shaping both localized orthopedic outcomes and systemic recovery trajectories.

### 4.1. Implications for Orthopedic Surgeons

#### 4.1.1. Preoperative Considerations

Orthopedic surgeons are increasingly recognizing the importance of systemic factors, including gut microbiota, in influencing surgical outcomes. It is worth considering that preoperative assessments should now evaluate gut microbiota status, as dysbiosis—an imbalance in gut bacteria—has been associated with systemic inflammation, impaired wound healing, and elevated surgical complication rates. Emerging tools, such as microbiota profiling and biomarkers like Zonulin and sCD14, can help identify patients at risk for dysbiosis-related complications. Elevated levels of these markers, along with inflammatory cytokines like interleukin-6 (IL-6) and C-reactive protein (CRP), signal gut barrier dysfunction and systemic inflammation, both of which are associated with poorer surgical outcomes [59,60].

Preoperative management of conditions linked to gut dysbiosis, such as diabetes and obesity, can reduce the risk of infections and complications. Studies show that dietary interventions, such as increasing prebiotic and probiotic intake, can improve gut microbiota composition and reduce systemic inflammation. Furthermore, antibiotics, while essential for infection prophylaxis, can disrupt the gut microbiota. Strategies like selective decontamination and the use of microbiome-sparing antibiotics may preserve microbial diversity while preventing surgical site infections. Emerging therapies, such as tailored probiotics, prebiotics, and synbiotics, show promise in enhancing gut health prior to surgery. For example, *Lactobacillus rhamnosus* and *Bifidobacterium longum* have demonstrated efficacy in reducing systemic inflammation and promoting mucosal barrier integrity [37].

#### 4.1.2. Postoperative Considerations

Postoperative outcomes in orthopedic surgery are significantly influenced by the interplay between systemic inflammation, immune response, and gut microbiota. Dysbiosis and gut barrier dysfunction are linked to higher rates of SSIs and PJIs. Studies highlight the importance of maintaining gut integrity to reduce bacterial translocation and systemic inflammation. Elevated Trimethylamine N-oxide (TMAO) levels and microbial endotoxins are correlated with delayed wound healing and implant failures [41,59].

Short-chain fatty acids (SCFAs), such as butyrate, produced by gut bacteria, are critical for reducing inflammation and promoting tissue repair. Enhancing SCFA production through dietary or probiotic interventions, post-surgery may accelerate recovery and improve bone healing outcomes. Dysbiosis has been associated with nonunion fractures and lower bone mineral density. Gut microbiota interventions, such as targeting SCFA-producing bacteria and mitigating inflammation, could enhance osteoblast activity and extend prosthetic longevity [39,43].

Fecal microbiota transplantation (FMT) is an emerging, microbiome-restorative therapy with potential relevance to orthopedic surgery. While clinical studies in this population are limited, preclinical evidence suggests that FMT may enhance bone health and reduce systemic inflammation. Ma et al. demonstrated that FMT in aged rats improved gut barrier integrity, restored microbial diversity, and mitigated bone loss [44]. FMT may hold promise for high-risk orthopedic patients with antibiotic-associated dysbiosis or persistent inflammation. Further investigation is warranted to assess its safety, feasibility, and efficacy in surgical settings. Postoperative therapies, such as FMT and tailored probiotics, are gaining traction. These interventions have the potential to mitigate dysbiosis, reduce systemic inflammation, and lower the risk of complications like PJIs and SSIs [44].

As the evidence base continues to grow, several microbiome-targeted strategies can be implemented in recovery, reduce systemic inflammation, and minimize complications. These include probiotics, prebiotics, microbiome testing, and selective antibiotic use. Table 4 summarizes these key interventions, their clinical rationale, recommended applications, and available tools for integration into surgical care.

#### 4.1.3. Microbiome Testing Protocols: Current Practices and Future Directions

PJIs and other SSIs are devastating complications in orthopedics as they may lead to prolonged treatments and revision surgeries. One major risk factor includes microbial colonization, but current practices to detect and address this vary. There is therefore a need for standardized microbial screening protocols conducted before orthopedic procedures in order to identify high-risk patients and implement targeted measures. Screening and decolonizing *S. aureus* carriers with mupirocin and chlorhexidine halved hospital-acquired infections (3.4% vs. 7.7%, RR 0.42). Additionally, routine *S. aureus* screening and decolonization significantly reduced SSIs and PJIs in joint arthroplasty patients [59]. Despite these advantages, these practices are not universal, so formalizing protocols could ultimately reduce these risk factors. Postoperative surveillance in high-risk patients is also essential. Closer monitoring through wound cultures or inflammatory markers can facilitate prompt detection and treatment of infections.

Orthopedic surgery can utilize microbial risk management protocols already present in other surgical fields. When considering solid organ transplant programs, there is a rigorous infection screening of donors and recipients as part of preoperative workup. This is attributed to checking for pathogens and colonization with multidrug-resistant organisms. Within cardiothoracic surgery, patients are often screened for *S. aureus* carriage prior to procedures such as the implantation of cardiac devices to reduce the risk of endocarditis and deep sternal wounds [61]. Colorectal surgery protocols also highlight the vital role of the microbiome as colorectal surgeons perform preoperative bowel decontamination to reduce intraoperative contamination. Across transplant, cardiac, and colorectal domains, microbiological screening and measures have been conducted to reduce infection risks, and orthopedic surgery can also benefit from a similar approach [62].

The growing interest in microbial testing in orthopedic surgery reflects a broader movement across medicine to establish formal guidelines for perioperative microbiome management. In recent years, leading professional societies in infectious disease and surgery have issued recommendations addressing the *Staphylococcus aureus* screening, antibiotic stewardship, and the management of patients colonized with multidrug-resistant organisms. For example, updated guidelines from the American Society of Health-System Pharmacists (ASHP) recommend adding vancomycin prophylaxis for patients known to carry MRSA or those deemed high-risk. Similarly, the International Consensus Group on Orthopedic Infections reached 85% agreement that nasal screening and decolonization should be standard practice to reduce periprosthetic joint infections (PJIs), an approach already adopted in many hospital protocols [63].

Research efforts are also underway to develop tools such as a microbiome risk index for use in preoperative assessments, and pilot programs are evaluating the integration of gut microbiome analysis into enhanced recovery pathways. As these initiatives evolve, more detailed guidelines tailored specifically to orthopedic surgery will likely emerge. A future framework could include mandatory *S. aureus* screening and decolonization for implant-based procedures, targeted prophylaxis for patients colonized with resistant organisms, and microbiome-based optimization strategies in high-risk cases. Although comprehensive guidelines are still in development, orthopedic surgery is shifting toward a preventive, microbiome-informed model. Implementing standardized microbial screening and surveillance now positions orthopedic teams to improve surgical outcomes and align with evolving clinical best practices [64].

Although formal microbiome protocols in orthopedic surgery are still emerging, the field is steadily moving toward a microbiome-conscious model. Incorporating standardized screening and surveillance practices positions orthopedic teams to improve surgical outcomes. To place this shift in a broader context, Table 5 draws on examples from general surgery, where microbiome-driven strategies have already shown clinical benefit. These cross-disciplinary insights reinforce the relevance of microbiome management beyond orthopedics and offer a practical framework for guiding future protocol development.

### 4.2. Study Strengths and Limitations

This review emphasizes the relevance of the gut microbiome in orthopedic surgical outcomes and examines its role in postoperative recovery, infection risk, and inflammatory responses. The synthesis of current literature indicates that shifts in microbial composition may influence surgical success. Tailored probiotic interventions may affect microbiome profiles, and predictive markers could help assess the risk of surgical complications. Preoperative optimization protocols may also be used to reduce the risk of adverse outcomes. Although the field is still maturing, the evidence points toward a future where microbiome-informed strategies may help reduce complications and enhance surgical recovery.

A strength of this review is its interdisciplinary approach by drawing from both surgical literature and microbiome research to identify patterns that may be overlooked when the two are studied in isolation. This study offers a foundation for hypothesis generation and future clinical investigation. It also highlights underexplored areas. such as cognitive dysfunction and the inflammatory cascade, where microbiome-targeted strategies hold promise.

Despite its strengths, this study faces limitations that reflect the nature of systematic reviews. Much of the available literature consists of observational studies, case series, and retrospective cohort analyses, which introduce potential biases and limit the ability to draw firm causal relationships. Many of the studies reviewed did not include large patient populations or randomized controlled trials (RCTs), which are the gold standard for clinical research. In addition, publication bias is also a concern in any systematic review, especially in fields where negative studies may remain unpublished.

The literature also demonstrates considerable methodological heterogeneity. Differences in sequencing platforms, sample processing, taxonomic classification, and reference databases may make direct comparisons between studies difficult. The lack of standardization makes it challenging to determine which findings are clinically meaningful. Furthermore, the association between microbiome composition and surgical outcomes may be confounded by the wide variations in patient populations. Some studies focus on young, relatively healthy individuals undergoing elective procedures, while others include older patients with pre-existing dysbiosis due to chronic illness or diet.

The timing of microbiome assessments also represents a limitation. Most studies assess the gut microbiota at a single time point—often preoperatively or within a short postoperative window. However, the microbiome is highly dynamic and can undergo significant shifts due to perioperative factors such as anesthesia, antibiotic administration, dietary changes, and hospitalization. Longitudinal studies tracking microbiome composition across multiple perioperative time points are necessary to better understand its role in recovery and complications. This is especially important as the literature on microbiome-targeted interventions remains in its early stages. Similarly, very few studies have tested direct interventions, such as the administration of probiotics, in orthopedic populations. Although these interventions are well-explored in fields like gastroenterology, their translation to surgical care remains limited.

### 4.3. Future Directions

Advancing microbiome research in orthopedic surgery will require a shift from observational associations to well-designed clinical trials. Most existing studies are descriptive or retrospective, and this limits their ability to guide practice. Further work should prioritize randomized controlled trials evaluating direct microbiome targeted interventions, such as preoperative probiotics or dietary modification. These trials should also consider the patient subgroups more likely to benefit, including those with high antibiotic exposure or baseline dysbiosis.

More rigorous and standardized methodologies should also be adopted. Current studies assess microbiome composition at a single perioperative time point. This fails to capture the dynamic changes that occur in response to surgery, anesthesia, antibiotics, and hospitalization. Longitudinal designs that track microbial shifts from baseline through recovery are essential to understanding causality. Methodological inconsistencies, such as variation in sequencing platforms, sample processing, and taxonomic pipelines, continue to limit comparability across studies. Establishing consensus protocols for sample collection, analysis, and interpretation will be critical for ensuring reproducibility and advancing clinical integration.

Research should ultimately account for patient-level variability that influences both microbiome composition and response to intervention. Age, comorbidities, nutrition, and prior antibiotic use all affect microbial resilience. Stratified study designs can help clarify who benefits most and under what conditions. With interdisciplinary collaboration and continued advances in sequencing and computational tools, the microbiome may emerge as a modifiable factor in surgical optimization, helping to bridge basic science and clinical care in meaningful and measurable ways.

## 5. Conclusions

Gut microbiota has influence in orthopedic surgical outcomes and is worth further exploration for possible interventions to improve care. The gut microbiota is one of the critical dynamic variables in the outcomes of orthopedic surgeries. However, microbial composition influences not only inflammation and healing but also recovery trajectories. Targeted modulation is essential in understanding the potential utility of interventions, especially probiotics, in mitigating surgical risk and improving recovery. With the evolution of orthopedic surgery, microbiological insights into the perioperative care of patients may allow personalized and precision medicine approaches aimed at optimizing patient recovery and minimizing surgical complications.

## Figures and Tables

**Figure 1 microorganisms-13-01048-f001:**
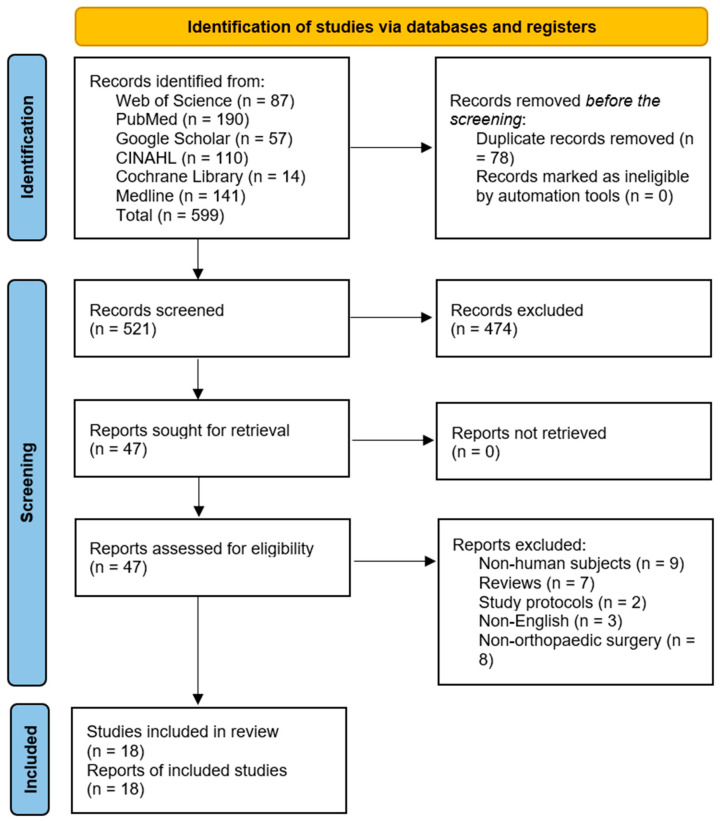
Preferred Reporting Items for Systematic Reviews and Meta-Analysis (PRISMA) flow diagram [26].

**Figure 2 microorganisms-13-01048-f002:**
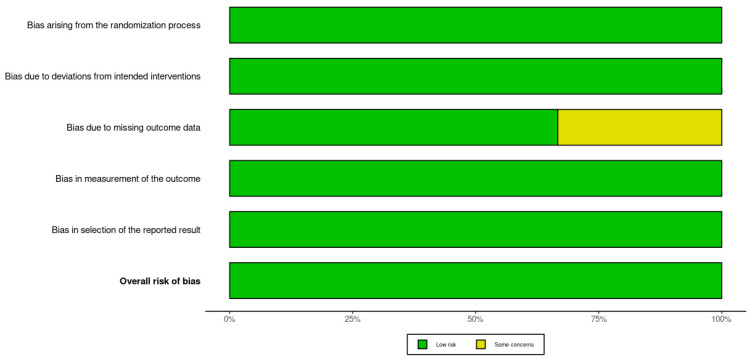
Traffic light plot of the Rob 2.0 assessment results [30,31,32].

**Figure 3 microorganisms-13-01048-f003:**
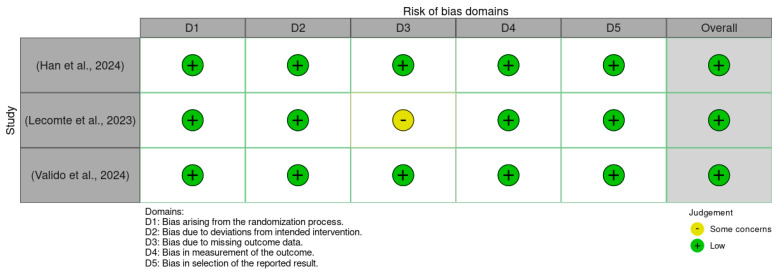
Summary plot of the Rob 2.0 assessment results for included studies [30,31,32].

**Figure 4 microorganisms-13-01048-f004:**
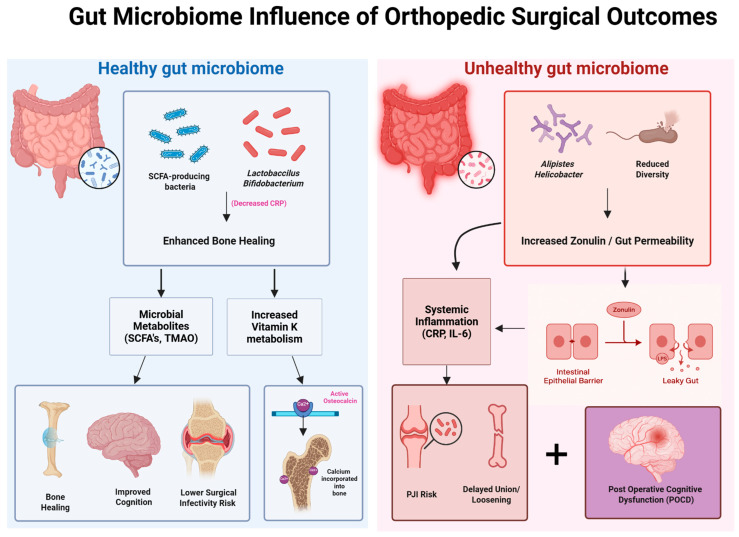
Impact of gut microbiome composition on orthopedic surgical outcomes. This schematic contrasts the effects of a healthy versus unhealthy gut microbiome on postoperative recovery. A healthy microbiome, rich in SCFA-producing bacteria and Lactobacillus/Bifidobacterium species, promotes enhanced bone healing through microbial metabolites (SCFAs, TMAO) and increased vitamin K metabolism, resulting in improved cognition, reduced infection risk, and better graft incorporation. In contrast, an unhealthy microbiome—marked by reduced diversity and abundance of bacteria such as Alistipes and Helicobacter—leads to increased zonulin expression and gut permeability, driving systemic inflammation (elevated CRP, IL-6), delayed osseous healing, heightened risk of periprosthetic joint infection (PJI), and postoperative cognitive dysfunction (POCD).

**Table 1 microorganisms-13-01048-t001:** Newcastle Ottawa Scale (NOS) assessment results for included studies [33,34,35,36,37,38,39,40,41,42,43,44,45].

Author	Selection		Comparability	Outcome	
	Representativeness of Cohort	Ascertainment of Exposure	Comparability of Cohorts	Assessment of Outcome	Follow Up
(Aboushaal a et al., 2024) [33]	*	*	*	*	*
(Chisari et al., 2022) [34]	*	*	*	*	*
(Ganta et al., 2023) [35]	*	*	*	*	*
(Kong et al., 2023) [36]	*	*	*	*	*
(Li et al., 2023) [37]	*	*	*	*	*
(Liu et al., 2022) [38]	*	*	*	*	*
(Cyphert et al., 2023) [39]	*	*	*	*	*
(Lin et al., 2020) [40]	*	*	*	*	*
(Li et al., 2024) [41]	*	*	*	*	*
(Baek et al., 2023) [42]	*	*	*	*	*
(Das et al., 2019) [43]	*	*	*	*	*
(Ma et al., 2021) [44]	*	*	*	*	*

An asterisk (*) indicates that the study fulfilled the corresponding criterion on the Newcastle–Ottawa Scale.

**Table 2 microorganisms-13-01048-t002:** National Institutes of Health (NIH) assessment results.

Study	Was the Research Objective Clearly Stated?	Was the Study Population Clearly Specified and Defined?	Was the Participation Rate of Eligible Persons at Least 50%?	Were All the Subjects Selected or Recruited from the Same or Similar Populations?	Were Effect Estimates Provided?	Were the Outcome Measures Clearly Defined, Valid, Reliable, and Implemented Consistently Across All Study Participants?	Were the Outcome Assessors Blinded to the Exposure Status of Participants?	Were Key Potential Confounding Variables Measured and Adjusted Statistically for Their Impact on the Relationship between Exposures and Outcomes?	Quality Rating
(Duan et al., 2021) [45]	Yes	Yes	Yes	Yes	Yes	Yes	Not mentioned	Not mentioned	Good
(Bi et al., 2023) [46]	Yes	Yes	Yes	Yes	Yes	Yes	Not mentioned	Not mentioned	Good
(Rosello-A non et al., 2023) [47]	Yes	Yes	Yes	Yes	Yes	Yes	Not mentioned	Not mentioned	Good

**Table 3 microorganisms-13-01048-t003:** Study characteristics.

Authors	Region	Sample Size, Demographics	Study Design	Intervention	Primary Outcomes: Infection Rates, Bone Healing	Secondary Outcomes: Pain, Inflammation, Hospital Stay	Objectives	Findings
(Abousha ala et al., 2024) [33]	USA	33 adults (21 with lumbar degenerative spondylolisthesis (LDS) and 12 without LDS), ages 18–80 years	Prospective cohort study	Observational study assessing gut microbiota and spinal health	LDS group shows higher disc degeneration severity (*p* = 0.018) and altered microbial diversity (*p* = 0.002–0.0046), with elevated proinflammatory bacteria (Dialister, CAG0352) and reduced anti-inflammatory bacteria (Slackia, Escherichia-Shigella)	No significant differences in pain profiles	To assess the association between gut microbiota dysbiosis and LDS in symptomatic patients.	Gut microbiome dysbiosis is significantly associated with LDS, characterized by higher Firmicutes-to-Bacteroidota ratio (*p* = 0.003) and shifts in pro-and anti-inflammatory bacterial taxa.
(Bi et al., 2023) [46]	China	40 elderly orthopedic patients	Prospective case-control study	Gut microbiota determined by 16S rRNA MiSeq sequencing	Preoperative gut microbiota composition	Neuropsychological assessments, postoperative pain scores, inflammatory markers, and hospital stay duration	Investigate the role of gut microbiota and metabolites in POCD in elderly orthopedic patients.	Preoperative gut microbiota differences identified in POCD patients; diagnostic efficiency demonstrated in 6 genera via ROC analysis; specific pathways impacting cognition function enriched.
(Chisari et al., 2022) [34]	USA	134 patients	Prospective cohort study	Zonulin, sCD14, LPS	Infection Rates: 44 PJI cases identified	Pain, Inflammation	Investigate the relationship between gut permeability and periprosthetic joint infection (PJI) through Zonulin, sCD14, and LPS biomarkers.	Zonulin and sCD14 were significantly elevated in PJI patients compared to non-infected. Acute PJI showed higher Zonulin levels than chronic PJI.
(Duan et al., 2021) [45]	China	80 patients	Case-control study	Intestinal microbiota using 16S rRNA	IM dysbiosis and barrier dysfunction were observed preoperatively in SCD and aMCI patients; elevated plasma LPS and CRP in SCD patients	Reduced levels of short-chain fatty acids-producing bacteria	Investigate preoperative differences in intestinal microbiota and barrier function in NC, SCD, and aMCI patients.	SCD patients had a lower Chao1 index, higher Bacteroidetes, and elevated plasma LPS/CRP. -aMCI patients exhibited increased Firmicutes and plasma occludin.
(Ganta et al., 2023) [35]	USA	551 adult patients	Retrospective cohort analysis	Culture-directed antibiotic treatment.	Time to fracture union: longer in gram-negative (662.3 days vs. 446.8 in gram-positive)	Number of reoperations, reconstructive surgeries	Assess bacterial microbiome affecting infected nonunions; Evaluate effects of bacterial speciation on fracture healing outcomes.	Gram-negative infections were associated with delayed union compared to gram-positive, though not statistically significant. *Staphylococcus* species were the most common isolates; reoperation rates were similar between groups.
(Kong et al., 2023) [36]	China	21: 11-spinal cord injury (SCI) and 10-healthy controls	Observational cohort study	Serum metabolites		Gut dysbiosis correlated with metabolic disorders, injury duration, and motor dysfunction severity	To examine changes in gut microbiota and metabolites in SCI patients. Analyze correlations between gut microbiota, metabolites, and clinical parameters.	SCI patients exhibited gut dysbiosis and distinct serum metabolite profiles. Microbial shifts correlated with neurological grade and injury duration.
(Li et al., 2023) [37]	China	77 patients	Prospective cohort study	Leukocyte esterase	Staphylococcus and Pseudomonas types showed higher inflammatory responses	Diagnostic value of microbiota typing for PJI confirmation	To characterize the periprosthetic microbiota in patients with suspected PJI.	Significant differences in periprosthetic microbiota between PJI and non-PJI groups.
(Liu et al., 2022) [38]	China	135 patients	Prospective observational cohort study	Measurement of gut microbiota, bacterial endotoxin, tight junction proteins. Knee arthroplasty or lumbar fusion.	Microbiota dysbiosis and intestinal barrier dysfunction	Increased level of inflammatory markers (CRP, IL-6, IL-10 levels monitored postoperatively)	To investigate changes in gut microbiota and intestinal barrier function after orthopedic surgery in patients with normal cognition vs. Alzheimer’s disease.	Gut dysbiosis linked with perioperative metabolic stress and inflammation.
(Cyphert et al., 2023) [39]	USA	31 patients undergoing spinal fusion surgery	Observational cohort study	Microbiome composition and bone mineral density (BMD).	Association between gut microbiota composition and bone mineral density (BMD) in spinal fusion patients	No secondary outcomes were specified	To investigate the relationship between microbiome composition and bone health in spinal fusion surgery patients.	A significant difference in microbiota composition between patients with low BMD (T-score ≤ −1.0) and those with normal BMD (*p* = 0.03). No significant changes in microbiome composition postoperatively.
(Han et al., 2024) [30]	China	100 fracture patients	RCT	Oral administration of Lactobacillus rhamnosus JYLR-127	Infection Rates: Decreased C-reactive protein level (*p* = 0.030)	Pain/Inflammation: Decreased abdominal PAC-SYM scores (*p* < 0.001); reduced inflammatory markers	To assess the efficacy of L. rhamnosus JYLR-127 in alleviating post-surgery constipation, reducing inflammation and improving gastrointestinal health.	Altered gut microbiota: Increased Firmicutes (*p* < 0.01), decreased Bacteroidetes (*p* < 0.05). Potential anti-infection and anti-inflammatory effects post-fracture surgery.
(Lecomte et al., 2023) [32]	Ireland	100 postmenopausal women	RCT	8-prenylarin genin (8-PN) and calcium and vitamin D3 supplements	Bone Healing/BMD: Total body BMD increased significantly in the HE group (1.8 ± 0.4% vs. baseline, *p* < 0.0001). The proportion of women with ≥ 1% BMD increases higher in the HE group (OR: 2.41, *p* < 0.05)	Quality of Life: Improvement in SF-36 physical functioning score (*p* = 0.05)	To assess the effect of an 8-PN standardized hop extract on bone status and understand the role of gut microbiome in postmenopausal osteopenic women.	HE supplementation increased total body BMD, improved physical functioning scores, and altered the abundance of gut microbiota genera.
(Lin et al., 2020) [40]	China	20 patients	Observational cohort study	Serum trimethylamine N-oxide	Elevated trimethylamine N-oxide levels in OP; promoted BMSCs	Increased ROS, IL-1β, IL-6, TNF-α; inhibited BMSCs proliferation activation of NF-κB pathway	To investigate the effect of trimethylamine N-oxide on bone metabolism in OP and elucidate mechanisms underlying these effects.	Elevated trimethylamine N-oxide negatively correlated with BMD in OP; NF-κB activation linked to trimethylamine N-oxide-induced ROS, inflammation, and altered BMSCs differentiation leading to bone loss.
(Li et al., 2024) [41]	China	126 patients	Retrospective cohort study	BMD assessment by DXA scan	OPN group showed elevated serum β-CTX levels, reduced α diversity, altered GM composition	Increased Escherichia-Shigella and Faecalibacterium	To investigate the GM profile in AIS patients with differing BMD and investigate the association between GM, osteopenia, and bone turnover.	Altered GM profile in the OPN group, with reduced diversity and specific genus-level changes; Escherichia-Shigella negatively correlated with femur BMD and positively with β-CTX.
(Valido et al., 2024) [31]	Switzerland	14	RCT	Multispecies-multistrain probiotic or prebiotic	Decrease inflammatory markers: probiotics led to a decrease in 83% (25/30) of inflammatory markers compared to prebiotics. Increased gut microbiome alpha diversity (Chao1 index higher post-probiotic use)	Increased GIQLI scores post-probiotic (not significant). Enterococcus faecium W54 increased after probiotic use	Investigate the effects of probiotics vs. prebiotics on inflammatory status and gut microbiome composition.	Probiotics showed the potential to reduce inflammation and improve gut microbiome diversity more effectively than prebiotics.
(Baek et al., 2023) [42]	Republic of Korea	37 patients (age: 62–83 years)	Prospective cohort study	Diagnostic analysis using α-defensive (AD), leukocyte esterase (LE) and metagenomic sequencing	Infection Rates 48.6% (18/37) classified as prosthetic joint infection (PJI)	Pain and Inflammation: The PJI group had higher ESR, CRP, synovial WBC, and PMN levels	To evaluate diagnostic biomarkers for PJI and investigate microbiome roles in synovial fluid.	PJI is associated with reduced microbial richness and pro-inflammatory taxa.
(Das et al., 2019) [43]	Ireland	181 participants	Retrospective cohort study	16S rRNA (V3–V4 region) amplicon sequencing of fecal microbiota	Bone Homeostasis: Altered microbiota associated with reduced bone mineral density (BMD)	Inflammation indirectly inferred via altered microbiota composition	Investigate the relationship between gut microbiota, bone homeostasis, and fracture risk in older adults.	BMD reduction is significantly associated with specific microbiota changes.
(Ma et al., 2024) [44]	USA	770,075 patients	Retrospective cohort study	Exposure history of *C. difficile* infection within 2 years before THA	Infection Rates: Higher odds of PJI within 2 years post-THA associated with prior *C. difficile* infection. Risk increased with closer proximity of infection to THA	Not directly assessed	To evaluate the effect of prior *C. difficile* infection on the risk of PJI following THA and explore proximity-related risk variations.	Prior *C. difficile* infection is an independent risk factor for PJI (OR: 1.49%, 95% CI: 1.09–2.02). -Closer proximity of C. difficile infection to THA increases PJI risk.
(Roselló-Añón et al., 2023) [47]	Spain	50 elderly patients	Case-control study	Analysis of gut	Elevated levels of Bacteroidales (*p* < 0.001) and Peptostreptococcales-Tissierellales (*p* < 0.005) in fracture patients	Alpha diversity indicated higher estimators at the taxonomic class level in fracture patients	To identify associations between gut microbiota composition and fragility of hip fractures in elderly patients.	Specific microbiota patterns were associated with the fragility of hip fractures.

**Table 4 microorganisms-13-01048-t004:** Evidence-based microbiome-targeted strategies in orthopedic surgery: clinical rationale, recommendations, and tools.

Evidence-Based Surgical Interventions	Rationale	Clinical Recommendations	Specific Products/Tests
Probiotic Supplementation	Certain probiotics (*Lactobacillus rhamnosus*, *Bifidobacterium longum*, *Lactobacillus acidophilus*), cognitive dysfunction, systemic inflammation, and surgical site infections.	Use probiotics preoperatively and postoperatively to support gut microbial balance and reduce infection risks. Consider probiotic supplementation in patients receiving broad-spectrum antibiotics to mitigate microbiome disruption. Monitor and educate patients on probiotic use before and after surgery.	(*Lactobacillus rhamnosus* GG), (*Bifidobacterium longum*), *Saccharomyhces boulardii*
Prebiotic Supplementation and Fiber-Rich Diets	Prebiotic intake (fermentable fibers) supports SCFA production, which enhances gut barrier integrity and reduces systemic inflammation.	Recommend a high-fiber, prebiotic-rich diet (<30 g/day) at least 2 weeks before surgery Encourage consumption of fermented foods (yogurt, kefir, kimchi, sauerkraut) for microbiome diversity Increase fiber intake through plant-based sources (legumes, whole grains, flaxseeds, leafy greens).	Inulin, Psyllium Husk, Fructooligosaccharides (FOS), Partially Hydrolyzed Guar Gum
Microbiome Testing	Testing for markers such as zonulin and soluble CD14 provides insight into gut permeability and inflammation.	Utilize microbiome sequencing to evaluate microbial diversity in high-risk patients. Screen at-risk patients for microbiome-related complications, such as those with recurrent infections, or gastrointestinal issues.	Microbiome sequencing tests Biomarkers: Zonulin (gut permeability), Soluble CD14 (systemic inflammation), Calprotectin (gut inflammation)
Targeted Antibiotic Use	Selective use of antibiotics can minimize microbiome disruption and preserve beneficial bacterial populations while preventing infections.	Use narrow-spectrum antibiotics when clinically appropriate to preserve gut microbiota. Consider microbiome-sparing antibiotics for prolonged perioperative prophylaxis. Monitor patients for antibiotic-induced dysbiosis and recommend probiotic co-administration where indicated.	Various microbiome sparing antibiotics
Gut Barrier Protection Strategies	Strategies like perioperative enteral nutrition help maintain gut integrity and prevent bacterial translocation, reducing infection risks.	Avoid prolonged fasting and ensure early enteral nutrition in critically ill patients. Reduce reliance on total parenteral nutrition when possible to maintain gut microbiome stability. Encourage hydration and electrolyte balance to support intestinal barrier function.	Immunonutrition Formulas (arginine, omega-3s, nucleotides)

**Table 5 microorganisms-13-01048-t005:** Impact of Microbiome Composition on Surgical Outcomes in Orthopedic and General Surgery.

Authors	Study	Year	Surgical Context	Microbiome Focus	Main Findings	Clinical Relevance
(Wei et al., 2024) [61]	Preoperative gut microbiota of POCD patients induces pre- and postoperative cognitive impairment and systemic inflammation in rats	2024	Orthopaedic surgery	Analyzing the preoperative gut microbiota of patients with postoperative cognitive dysfunction (POCD).	The gut microbiota of POCD patients exhibited dysbiosis preoperatively	Targeting the microbiota preoperatively could improve postoperative cognitive function.
(Guerrero et al., 2021) [65]	Adherence to a standardized infection reduction bundle decreases surgical site infections after colon surgery: a retrospective cohort study on 526 patients	2016–2017	Colon Surgery	Emphasis on minimizing colon surgical site infections (SSI), indirectly impacting microbiome balance.	Implementing a perioperative SSI prevention bundle lowered SSI rates from 8.7% to 1.2%. Standard infection rate was reduced by 85.4% after introducing the prevention bundle	Advocates for implementing standardized SSI prevention bundles to improve patient safety and decrease infection rates in colon surgery.
(Song et al., 2018) [66]	Bacterial culture and antibiotic susceptibility in patients with acute appendicitis	2006–2015	Appendectomy for acute appendicitis	Detection of pathogens in appendicitis and their antibiotic resistance profiles.	*Escherichia coli* (64.6%) and *Pseudomonas aeruginosa* (16.4%) were the most common microorganisms for SSI. *Pseudomonas aeruginosa* was a significant microorganism associated with SSI (OR = 2.128, 95% CI = 1.077–4.206, *p* = 0.030)	Emphasizes the need for proper empirical antibiotic selection, particularly for *P. aeruginosa* infections, to minimize postoperative complications.
(Effenberger et al., 2023) [67]	Transmission of oral microbiota to the biliary tract during endoscopic retrograde cholangiography	2010–2011	Endoscopic retrograde cholangiography (ERC)	Microbial contamination of bile from the oral cavity and endoscopic equipment.	*Bacteroides fragilis* was significantly associated with cholangitis (*p* = 0.015). Microbial translocation from the throat and endoscope to bile was common, clinical outcomes not affected	Microbial contamination during ERC is frequent; it does not necessarily lead to adverse patient outcomes.
(Khoury et al., 2024) [68]	Gastrointestinal endoscopy 30-day-associated bacteremia: Nonoutbreak 5-year review in an inner-city, tertiary-care hospital	2018–2022	Gastrointestinal endoscopic procedures (GIEPs)	Incidence and associated factors of bacteremia associated with GIEPs.	Bacteremia rate was highest after endoscopic retrograde cholangiopancreatography (2.84%) and lowest after colonoscopy (0.08%)	Emphasizes the importance of clinical surveillance to reduce infection risk after GIEPs.

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
