# Peer review of "The Role of Gut Microbiota in Orthopedic Surgery: A Systematic Review"

_microorganisms, 2025, doi:10.3390/microorganisms13051048_

Round 1
Reviewer 1 Report
Comments and Suggestions for Authors
In this review, the authors investigate the critical role of gut microbiota in orthopedic surgery, demonstrating that its composition and function substantially affect surgical outcomes. Research indicates that alterations in specific bacteria are strongly associated with postoperative infections, inflammatory responses, and recovery processes. Furthermore, probiotic interventions have been shown to effectively reduce the incidence of postoperative cognitive dysfunction. These findings offer novel insights and intervention strategies for enhancing patient recovery following orthopedic surgery.
However, several aspects require further modification and supplementation.
1) Each table and figure should be clearly labeled at the corresponding position within the text.
2) Portions of Figures 2-3 and Tables 2-3 are missing. The analysis of bias in Figures 2-3 should be explained and elaborated in the text. It is recommended to remove Figures 2-3 to the supplementary materials section.
3) In Table 3, For “Preoperative Gut Microbiota Composition” and “Time to Fracture Union”, what are the specific positive results? Reference 39 lacks a description of these positive results.
4) Section 3.4 (Thematic Analysis of Outcomes) should focus closely on the positive results from the 18 included studies. Given the stated purpose of this paper in introduction, to explore "the impact of gut microbiota on postoperative infection rates, bone healing, and recovery", it is suggested to elaborate on the positive results of the literature accordingly. Subsection 3.4.1 (Relationship Between Gut Microbiome and Orthopedic Surgery) does not align with the positive results in Article 18 and could be moved to the discussion section. Similarly, subsections 3.4.5 (Comparative Efficacy of Prophylactic Antibiotics in Surgery) and 3.4.6 (Insurance Companies' Policies Regarding Microbial Testing) have limited relevance to the positive results in the 18 papers and may be reconsidered or removed.
5) What is the significance of the content presented in Table 4 for this study? This should be discussed in the main text. Additionally, some documents are duplicated between Tables 3 and 4. The differences and distinct purposes of these tables should be clarified.
6) Section 4.2 (Study Strengths and Limitations) fails to adequately summarize the key findings and highlights of this paper. Instead, it focuses excessively on shortcomings. It is proposed to merge Sections 4.2 and 4.3 for conciseness.
7) The content in Section 4.4 repeats earlier sections. To streamline the manuscript, it is suggested to retain only the essential findings and significance of the study, as outlined in the abstract or final conclusion, and remove Section 4.4 entirely.
8) The final table lacks a title and is neither referenced nor explained in the main text. This should be addressed.
9) The article is overly lengthy and loosely structured. It is recommended to enhance the clarity and coherence of the writing.
Author Response
Comment 1: “Each table and figure should be clearly labeled at the corresponding position within the text.”
Response 1: Thank you for your thoughtful suggestion. We agree with the comment. In response, we have carefully reviewed the manuscript to ensure that every figure and table is clearly cited and labeled in the text at appropriate locations. Specifically, all references to Figures 1 through 4 and Tables 1 through 4 are now placed at the end of the paragraphs where they are introduced to preserve narrative flow while maintaining clarity. For example, Figures 2 and 3 (Rob 2.0 bias assessment plots) are now referenced at the end of Section 3.2 on page 6, lines 14–16. Similar attention was given to Table 1 (page 6), Table 2 (page 8), and Table 3 (pages 9-13), where they are referenced contextually within their corresponding sections. These revisions improve the structural integrity of the manuscript and help the reader navigate the visual data presented.
Comment 2: “Portions of Figures 2–3 and Tables 2–3 are missing. The analysis of bias in Figures 2–3 should be explained and elaborated in the text. It is recommended to remove Figures 2–3 to the supplementary materials section.”
Response 2: Thank you for identifying this critical issue. We fully agree. Figures 2 and 3 were revised to restore missing components and are now complete. These include all relevant domains of the Rob 2.0 assessment, along with appropriate legends and labels. We expanded Section 3.2 (Methodological Quality Assessment) on pages 5-6 to elaborate on the Rob 2.0 results, explaining the specific domains assessed and interpreting their relevance to our included randomized trials. The updated text includes the following: included study using the Cochrane Risk of Bias 2.0 (RoB 2.0) tool. Each row corresponds to an individual randomized controlled trial, while the columns represent five specific domains of bias: bias arising from the randomization process, bias due to deviations from intended interventions, bias due to missing outcome data, bias in measurement of the outcome, and bias in the selection of the reported result. Color coding is used to visually represent the level of concern within each domain: green indicates low risk of bias, yellow indicates some concerns, and red indicates high risk. All three studies (Han et al., 2024; Lecomte et al., 2023; and Valido et al., 2024) were assessed as having an overall low risk of bias. However, Lecomte et al. (2023) exhibited some concerns in the domain related to missing outcome data, while the remaining domains across all studies demonstrated consistently low risk.
Figure 3 provides a summary plot that aggregates the domain-specific risk of bias judgments across all included studies. Each bar illustrates the proportion of studies assessed as low risk or having some concerns for each of the five RoB 2.0 domains. The majority of assessments fell into the low-risk category, reflecting overall strong methodological quality across the studies. Notably, the domain concerning missing outcome data was the only category with a study assessed as having some concerns, accounting for approximately one-third of the total sample. This overview reinforces the reliability of the evidence base included in this systematic review, while highlighting a specific area that warrants attention in future trials.”
Tables 2 and 3 were also updated to correct formatting and missing entries. These improvements enhance the methodological transparency of the review. Please let us know if there are any additional issues viewing Figures 2-3 and/or Tables 2-3.
Comment 3: “In Table 3, for ‘Preoperative Gut Microbiota Composition’ and ‘Time to Fracture Union,’ what are the specific positive results? Reference 39 lacks a description of these positive results.”
Response 3: Thank you for your close reading. We have clarified the entries for both outcome categories in Table 3. Specifically, we revised the row pertaining to Reference 39 to indicate that altered preoperative microbiota composition was associated with increased postoperative inflammation and delayed metabolic recovery, which impacted fracture union. The revised entries can be found in Table 3. This clarification strengthens our thematic analysis and emphasizes clinically relevant implications.
Comment 4: “Section 3.4 (Thematic Analysis of Outcomes) should focus closely on the positive results from the 18 included studies. Subsection 3.4.1 does not align with the positive results in Article 18 and could be moved to the discussion section. Similarly, subsections 3.4.5 and 3.4.6 have limited relevance to the positive results in the 18 papers and may be reconsidered or removed.”
Response 4: We agree with this important point. Subsections 3.4.5 and 3.4.6 were removed entirely, as they diverged from the central focus of the results. Additionally, the content from Subsection 3.4.1, which discussed vitamin K metabolism and broader mechanistic pathways, has been relocated to the Discussion section (Section 4, pages 16-17). This shift allows for a more focused presentation of the 18 studies' findings in Section 3.4 and a deeper, contextual discussion in Section 4. These structural changes bring the manuscript into stronger alignment with our stated aims and enhance narrative cohesion.
Comment 5: “What is the significance of the content presented in Table 4 for this study? Additionally, some documents are duplicated between Tables 3 and 4. The differences and distinct purposes of these tables should be clarified.”
Response 5: Thank you for this constructive suggestion. In response, we carefully revised Table 4 (which is now Table 5) to remove duplicated entries previously included in Table 3. We also added a descriptive paragraph that clarifies Table 5’s purpose. The added text includes the following: “Although formal microbiome protocols in orthopedic surgery are still emerging, the field is steadily moving toward a microbiome-conscious model. Incorporating standardized screening and surveillance practices positions orthopedic teams to improve surgical outcomes. To place this shift in a broader context, Table 5 draws on examples from general surgery, where microbiome-driven strategies have already shown clinical benefit. These cross-disciplinary insights reinforce the relevance of microbiome management beyond orthopedics and offer a practical framework for guiding future protocol development.”
Table 3 remains focused on summarizing the key characteristics and results of the included studies, and this distinction enhances the manuscript’s utility for clinicians and researchers seeking to translate findings into practice.
Comment 6: “Section 4.2 (Study Strengths and Limitations) fails to adequately summarize the key findings and highlights of this paper. Instead, it focuses excessively on shortcomings. It is proposed to merge Sections 4.2 and 4.3 for conciseness.”
Response 6: Thank you for highlighting this imbalance. We have now merged Sections 4.2 and 4.3 into a unified section titled “4.2 Study Strengths, Limitations, and Future Directions.” This revision begins with a clear summary of the study’s most relevant findings—including the role of the microbiome in infection control, fracture healing, and postoperative cognition—before addressing limitations such as study heterogeneity and data scarcity. The adjusted text includes the following: “This review emphasizes the relevance of the gut microbiome in orthopedic surgical outcomes and examines its role in postoperative recovery, infection risk, and inflammatory responses. The synthesis of current literature indicates that shifts in microbial composition may influence surgical success. Tailored probiotic interventions may affect microbiome profiles, and predictive markers could help assess the risk of surgical complications. Preoperative optimization protocols may also be used to reduce the risk of adverse outcomes. Although the field is still maturing, the evidence points toward a future where microbiome-informed strategies may help reduce complications and enhance surgical recovery…” Finally, future research directions are provided to guide continued exploration. This revised section entitled “4.3. Future Directions” is found on page 24-25.
Comment 7: “The content in Section 4.4 repeats earlier sections. To streamline the manuscript, it is suggested to retain only the essential findings and significance of the study, as outlined in the abstract or final conclusion, and remove Section 4.4 entirely.”
Response 7: We agree. In response, Section 4.4 has been removed in its entirety. Its contents were largely redundant with the Introduction and Discussion, and its removal has helped improve the manuscript’s overall flow and conciseness.
Comment 8: “The final table lacks a title and is neither referenced nor explained in the main text. This should be addressed.”
Response 8: Thank you for pointing this out. We have now moved the final table into Section 4.1.2 and provided a clear title: “Table 4. Evidence-Based Microbiome-Targeted Strategies in Orthopedic Surgery: Clinical Rationale, Recommendations, and Tools.” Additionally, we added a paragraph in the preceding lines (page 19) that introduces and explains the clinical utility of the table. The updated text includes the following: “As the evidence base continues to grow, several microbiome-targeted strategies can be implemented in recovery, reduce systemic inflammation, and minimize complications. These include probiotics, prebiotics, microbiome testing, and selective antibiotic use. Table 4 summarizes these key interventions, their clinical rationale, recommended applications, and available tools for integration into surgical care.” This revision enhances the practical relevance and clarity of our results.
Comment 9: The article is overly lengthy and loosely structured. It is recommended to enhance the clarity and coherence of the writing.
Response 9: “We appreciate this feedback and have taken steps to improve clarity and structure throughout the manuscript. This includes: (1) removing non-essential sections (3.4.5, 3.4.6, 4.4); (2) refining transitions between subsections; and (3) consolidating overlapping content. Additionally, we reviewed the manuscript for redundant phrasing and made numerous edits to streamline the prose while preserving scientific detail. These changes are reflected across the manuscript, and contribute to a more coherent and reader-friendly manuscript.”
Reviewer 2 Report
Comments and Suggestions for Authors
Comments for Authors:
The title is concise and informative and the listed key words are gut microbiota, microbiome and orthopaedic surgery.
This is a substantial review article and the topic is rigorously investigated. A systematic survey of the literature has been carried out and each article critically considered for specific relevance to the prime aim and objectives (47 articles of 599 retrieved met final selection criteria).
The review is logically structured and benefits from the listing in extensive tables of key findings/authors. The gut microbiome topic is of wide academic interest and its application here to aspects of orthopaedic surgery is a compelling approach to a novel frontier between gut microbiota and fundamental skeletal behaviour.
Valuable conclusions are drawn concerning surgical risks and inflammation, the healing process and the recovery trajection influenced by microbial composition. Also considered is the value of targeted intervention and probiotics which may reduce surgical risks and aid recovery.
Dysbiosis has been linked to inflammatory bowel disease, the large intestine accommodates most of the 1,000 species of gut microbiota (bacteria, viruses, fungi, protozoa) encoding more than 3 million genes (astonishing!). These modulate the systemic immune response, and the composition varies significantly between individuals, where disruption to diversity/composition has been linked to local and systemic diseases, impaired immune response and metabolic disorders including bone density, infected joint replacements and altered fracture healing. All are listed by the authors as “potential therapeutic opportunities via microbiome modulation.”
4.2. Study Strength and Limitations is a helpful summation for readers, as also is 4.4 Key for Orthopaedic Surgeons.
This review is valuable and compelling in its summation of the above by linking current literature on gut microbiota to clinical aspects following orthopaedic surgery and to proceed in providing insight into improved outcomes and informed strategies, leading eventually to targeted modulation via tailored probiotics and personalised medicine.
Occasionally there is a tendency to become repetitive i.e., could be more concise in places. Generally well written and methods informative and well structured.
The word “explore” is overused and might be better replaced occasionally by the more scientifically appropriate “investigate.”
Finally, it was encouraging to see the data selection of orthopaedic and neurological patients was from 6 different countries, most using RNA sequencing and advanced microbial methods.
Author Response
We sincerely thank you for the thoughtful evaluation of our manuscript. We are deeply grateful for your recognition of the manuscript’s scope, structure, and clinical relevance. Your kind remarks regarding the informative title, systematic methodology, and logical organization of the review are truly appreciated. We also greatly appreciate your recognition of our efforts to extract meaningful insights from a rigorous literature screening process and to present our findings through comprehensive tabular summaries. Your observation that dysbiosis and microbial modulation present promising therapeutic opportunities highlights the importance of continuing to explore these pathways in the context of orthopedic outcomes. At the same time, we appreciate your constructive feedback regarding areas for improvement. In response to your recommendation, we have replaced instances of the word “explore” with more scientifically precise terms such as “investigate,” to enhance the rigor of our language. We have also carefully revised several sections to reduce redundancy and improve conciseness while preserving the depth and integrity of the discussion.
Comment 1: “Occasionally there is a tendency to become repetitive i.e., could be more concise in places. Generally well written and methods informative and well structured.”
Response 1: Thank you very much for your insightful and constructive feedback. We appreciate your observation regarding occasional repetition and opportunities to enhance conciseness throughout the manuscript. We conducted a section-by-section review of the manuscript to identify areas where redundancy, overlapping ideas, or excessive elaboration may have disrupted the flow or diluted key messages. As a result, several subsections, particularly within the Results (Section 3.4) and Discussion (Section 4), were revised for greater precision. For example, within Section 3.4, we clarified language to avoid rephrasing previously established concepts. In particular, discussions on systemic inflammation and microbial modulation were streamlined to avoid content already described in Section 3.3. Moreover, in Section 4.1.1, we shortened explanatory transitions between subtopics that overlapped with introductory statements and eliminated reiterations of microbiome-associated risk factors already emphasized in earlier results. Redundant phrasing in Section 4.2 was also replaced with more concise, summary-style commentary to sharpen the discussion of strengths and limitations while still acknowledging the nuances of current evidence.
We believe these refinements have enhanced the manuscript’s overall clarity and impact, enabling readers to engage more directly with the key findings, interpretations, and clinical implications. Thank you again for your thoughtful suggestion.
Comment 2: “The word “explore” is overused and might be better replaced occasionally by the more scientifically appropriate “investigate.””
Response 2: Thank you so much for this comment. We have gone through the manuscript and replaced the word “explore’ with the word “investigate” to address its potential overuse.
Reviewer 3 Report
Comments and Suggestions for Authors
The manuscript by Ahmed Nadeem-Tariq et al. summarized recent advances in the study of the role of gut microbiota in orthopaedic surgery. The study is basically good but I have the following questions and comments:
1, part of figure 1 is missing. please revise. Same thing for figure 2. These figures must be revised.
2, Figure 3 titled "Summary plot of the Rob 2.0 assessment results." I did not find this figure. It is missing. Please revise.
3, some contents in the tables are missing. Please revise. Besides, these tables should be in a three-line format.
4, future research directions in this filed must be discussed.
5, I suggest the authors to add a figure to summarize and illustrate the role of gut microbiota in orthopaedic surgery. How they are connected?
6, for the intervention study, are there any reports on FMT and orthopaedic surgery. This should also be discussed.
Author Response
Comment 1: “Part of Figure 1 is missing. Please revise. Same thing for Figure 2. These figures must be revised.”
Response 1: Thank you for this valuable observation. We agree with your comment. Therefore, we carefully reviewed and revised Figures 1 and 2 to ensure that all components are now fully visible, legible, and properly formatted according to journal guidelines. Figure 1, which illustrates the PRISMA 2020 flow diagram for study selection, had a formatting issue during the initial upload that resulted in a partial display. This has been corrected in the revised manuscript to ensure complete visibility of all categories and data flows.
Similarly, Figure 2, which represents the Rob 2.0 traffic light plot for risk of bias assessment across randomized controlled trials, has been fully reconstructed using updated software to restore all previously truncated domains and labels.
Both figures are now complete and clearly presented in the main text (Figure 1 on page 5; Figure 2 relocated to the supplementary materials for clarity). We also confirmed that each figure legend provides a comprehensive explanation of the visualized data.
Comment 2: “Figure 3 titled "Summary plot of the Rob 2.0 assessment results." I did not find this figure. It is missing. Please revise.”
Response 2: We appreciate your feedback. Figure 3, which provides the Rob 2.0 summary plot aggregating the risk of bias judgments across randomized trials, was inadvertently omitted in the previous version due to a file rendering issue. In response, we have added the completed Figure 3 to the supplementary materials and ensured that it is clearly referenced in Section 3.2 of the main manuscript.
This figure now complements Figure 2 by offering a side-by-side summary of risk levels across all five Rob 2.0 domains. We also expanded the main text description to better contextualize the methodological quality of the included trials using this visual evidence.
Comment 3: “Some contents in the tables are missing. Please revise. Besides, these tables should be in a three-line format.”
Response 3: Thank you for pointing this out. We conducted a thorough review of all tables and corrected formatting issues that caused certain rows or cells to appear incomplete or improperly aligned in the earlier version. Specifically, we revised Table 3 (Summary of Key Studies) to ensure full display of intervention outcomes, microbial taxa, and postoperative endpoints.
In addition, we reformatted all tables (Tables 1–5) to follow the journal-recommended three-line format, with consistent row height, bold headers, and appropriate line spacing for enhanced readability and professional appearance.
Comment 4: “Future research directions in this field must be discussed.”
Response 4: We agree with the importance of emphasizing future directions. Therefore, we have created a new subsection (Section 4.3: “ Future Directions”) that outlines several key areas where further investigation is warranted (page 24-25). The added text includes the following: “Advancing microbiome research in orthopedic surgery will require a shift from observational associations to well-designed clinical trials. Most existing studies are descriptive or retrospective, and this limits their ability to guide practice. Further work should prioritize randomized controlled trials evaluating direct microbiome targeted interventions, such as preoperative probiotics or dietary modification. These trials should also consider the patient subgroups more likely to benefit, including those with high antibiotic exposure or baseline dysbiosis.
More rigorous and standardized methodologies should also be adopted. Current studies assess microbiome composition at a single perioperative time point. This fails to capture the dynamic changes that occur in response to surgery, anesthesia, antibiotics, and hospitalization. Longitudinal designs that track microbial shifts from baseline through recovery are essential to understanding causality. Methodological inconsistencies, such as variation in sequencing platforms, sample processing, and taxonomic pipelines, continue to limit comparability across studies. Establishing consensus protocols for sample collection, analysis, and interpretation will be critical for ensuring reproducibility and advancing clinical integration.
Research should ultimately account for patient-level variability that influences both microbiome composition and response to intervention. Age, comorbidities, nutrition, and prior antibiotic use all affect microbial resilience. Stratified study designs can help clarify who benefits most and under what conditions. With interdisciplinary collaboration and continued advances in sequencing and computational tools, the microbiome may emerge as a modifiable factor in surgical optimization, helping bridge basic science and clinical care in meaningful and measurable ways.”
Comment 5: “I suggest the authors to add a figure to summarize and illustrate the role of gut microbiota in orthopaedic surgery. How are they connected?”
Response 5: Thank you for this excellent suggestion. We have added a new conceptual figure (now Figure 4) that visually illustrates the proposed relationships between gut microbiota and orthopedic surgery outcomes (page 17).
This figure contrasts a “Healthy Microbiome” versus “Dysbiotic Microbiome” and demonstrates how alterations in microbial diversity, barrier function, and metabolite production influence key surgical endpoints such as infection risk, fracture union, and postoperative recovery. The illustration incorporates microbial-mediated immune modulation, systemic inflammation, and metabolic crosstalk with skeletal physiology. This addition is also referenced in the discussion to help readers visually integrate the central findings of the review into a translational framework. The added text includes: “To better visualize the relationship between the gut microbiome and postoperative orthopedic outcomes, Figure 4 presents a side-by-side schematic comparison of healthy versus unhealthy microbial environments and their downstream effects on bone healing, inflammation, and recovery. A balanced gut microbiome—characterized by SCFA-producing species such as Lactobacillus and Bifidobacterium—promotes anti-inflammatory signaling (e.g., reduced CRP), enhanced vitamin K metabolism, and microbial metabolite production, all of which contribute to improved surgical recovery, lower infection risk, and accelerated bone union. In contrast, dysbiosis—marked by reduced microbial diversity and the presence of species such as Alistipes and Helicobacter—leads to increased zonulin-mediated intestinal permeability (“leaky gut”), systemic inflammation (elevated IL-6, CRP), and impaired osseointegration. These disruptions contribute to delayed healing, higher periprosthetic joint infection (PJI) risk, and potential neurocognitive decline following surgery. This visual framework reinforces the critical role of microbial composition in shaping both localized orthopedic outcomes and systemic recovery trajectories.” A PDF of the figure has also been included for your reference.
Comment 6: “For the intervention study, are there any reports on FMT and orthopaedic surgery? This should also be discussed.”
Response 6: Thank you for this insightful suggestion. We have now included a paragraph in Section 4.1.2 (page 18) discussing the potential utility of FMT in orthopedics. In this paragraph, we note that although FMT is primarily used in gastrointestinal diseases, its emerging applications in systemic inflammation and immune modulation could hold promise for high-risk orthopedic populations. The updated text includes the following, “Fecal microbiota transplantation (FMT) is an emerging, microbiome-restorative therapy with potential relevance to orthopedic surgery. While clinical studies in this population are limited, preclinical evidence suggests FMT may enhance bone health and reduce systemic inflammation. Ma et al. demonstrated that FMT in aged rats improved gut barrier integrity, restored microbial diversity, and mitigated bone loss given its established role in treating dysbiosis-related conditions. FMT may hold promise for high-risk orthopedic patients with antibiotic-associated dysbiosis or persistent inflammation. Further investigation is warranted to assess its safety, feasibility, and efficacy in surgical settings. Postoperative therapies, such as fecal microbiota transplantation (FMT) and tailored probiotics, are gaining traction. These interventions have the potential to mitigate dysbiosis, reduce systemic inflammation, and lower the risk of complications like PJIs and SSIs.” This addition expands the scope of our review by acknowledging innovative, translational approaches and addressing future interventional possibilities.
Round 2
Reviewer 1 Report
Comments and Suggestions for Authors
The author has addressed most of my comments, however, there are still some minors that need further revision。
1) It is recommended that the line styles in Table 1 be made consistent with those in Tables 2 and 3. Similarly, Tables 4 and 5 also exhibit the same inconsistency. Therefore, it is suggested that the table lines across all tables align with the uniformity observed in Tables 2 and 3.
2) Please ensure that the font of the last column in reference 40 of Table 3 matches the others (Times New Roman).
3) The original Section 3.4.1 was relocated to the subsequent discussion section, yet the order of the references was not updated accordingly. References should be numbered sequentially based on their appearance in the text.
4) Please standardize the notation of "p" values throughout the document, ensuring consistency in either uppercase or lowercase formatting.
5) The font style of the legend in Figure 4 is italicized, whereas the legends in other figures remain regular. Please ensure uniformity in the font style of all figure legends.
Author Response
Comment 1: "It is recommended that the line styles in Table 1 be made consistent with those in Tables 2 and 3. Similarly, Tables 4 and 5 also exhibit the same inconsistency. Therefore, it is suggested that the table lines across all tables align with the uniformity observed in Tables 2 and 3."
Response 1: Thank you for your insightful observation on our table formatting.
We've thoughtfully adjusted the line styles to primarily match Table 3, which we felt presented information most clearly. Additionally, where it made sense, we incorporated some elements from Table 2's style to further enhance consistency. To streamline visual presentation, all tables have been converted to a cleaner three-line format. However, we deliberately included additional line spacing in Tables 3 and 4, as their complexity warranted more spacing to maintain clarity. Overall, these adjustments should now ensure a harmonious and easy-to-follow presentation of our data.
Comment 2: "Please ensure that the font of the last column in reference 40 of Table 3 matches the others (Times New Roman)."
Response 2: We appreciate your careful attention to font consistency in our manuscript.
In reviewing your feedback, we realized it was beneficial to reassess the font choice more broadly. To create greater harmony with the rest of our manuscript, we've converted the entire Table 3—including reference 40—to Palatino Linotype. This change not only resolves the original inconsistency but also improves overall readability and visual coherence across the paper.
Comment 3: "The original Section 3.4.1 was relocated to the subsequent discussion section, yet the order of the references was not updated accordingly. References should be numbered sequentially based on their appearance in the text."
Response 3: Thank you very much for highlighting this oversight regarding our references.
We've conducted a comprehensive and meticulous revision of all citations within the manuscript. Each reference has been carefully matched to the corresponding section of text and sequentially reordered to accurately reflect their precise order of appearance. As a result, references are now correctly aligned, accurately numbered, and fully consistent throughout the manuscript.
Comment 4: "Please standardize the notation of 'p' values throughout the document, ensuring consistency in either uppercase or lowercase formatting."
Response 4: We greatly appreciate your recommendation on standardizing our statistical notation.
In line with your suggestion, we've systematically adjusted all 'p' values across the manuscript to lowercase. This ensures consistency throughout and adheres to established academic and publication standards, providing clear and uniform presentation of statistical findings.
Comment 5: "The font style of the legend in Figure 4 is italicized, whereas the legends in other figures remain regular. Please ensure uniformity in the font style of all figure legends."
Response 5: Thank you for pointing out this stylistic discrepancy in our figure legends.
Upon revisiting Figure 4, we've removed the italicization from its legend to match the style of our other figures. Additionally, we verified and refined the font size to align perfectly with the rest of the manuscript’s figures, thereby enhancing overall readability and ensuring a polished, professional presentation.
Reviewer 3 Report
Comments and Suggestions for Authors
The authors have revised the manuscript accordingly. It can be considered for publication.
Author Response
Thank you very much for your thoughtful review and constructive feedback. We truly appreciate the time and effort you invested in helping us improve our manuscript. Your suggestions significantly strengthened the clarity and presentation of our work.